# Sound Sensing: Generative and Discriminant Model-Based Approaches to Bolt Loosening Detection

**DOI:** 10.3390/s24196447

**Published:** 2024-10-05

**Authors:** Liehai Cheng, Zhenli Zhang, Giuseppe Lacidogna, Xiao Wang, Mutian Jia, Zhitao Liu

**Affiliations:** 1Shandong Electric Power Engineering Consulting Institute Corp., Ltd., Jinan 250013, China; chengliehai@sdepci.com (L.C.); zhangzhenli@sdepci.com (Z.Z.); 2Department of Structural, Geotechnical and Building Engineering, Politecnico di Torino, 10129 Torino, Italy; giuseppe.lacidogna@polito.it; 3School of Civil Engineering, Tianjin University, Tianjin 300350, China; ethan@tju.edu.cn (X.W.); jiamutian4218@tju.edu.cn (M.J.); 4Institute of Ocean Energy and Intelligent Construction, Tianjin University of Technology, Tianjin 300384, China

**Keywords:** bolt loosening, mel frequency cepstrum coefficients (MFCCs), cumulative energy entropy (CEE), gaussian discriminant analysis (GDA), support vector machine (SVM)

## Abstract

The detection of bolt looseness is crucial to ensure the integrity and safety of bolted connection structures. Percussion-based bolt looseness detection provides a simple and cost-effective approach. However, this method has some inherent shortcomings that limit its application. For example, it highly depends on the inspector’s hearing and experience and is more easily affected by ambient noise. In this article, a whole set of signal processing procedures are proposed and a new kind of damage index vector is constructed to strengthen the reliability and robustness of this method. Firstly, a series of audio signal preprocessing algorithms including denoising, segmenting, and smooth filtering are performed in the raw audio signal. Then, the cumulative energy entropy (CEE) and mel frequency cepstrum coefficients (MFCCs) are utilized to extract damage index vectors, which are used as input vectors for generative and discriminative classifier models (Gaussian discriminant analysis and support vector machine), respectively. Finally, multiple repeated experiments are conducted to verify the effectiveness of the proposed method and its ability to detect the bolt looseness in terms of audio signal. The testing accuracy of the trained model approaches 90% and 96.7% under different combinations of torque levels, respectively.

## 1. Introduction

As one of the key components of common building blocks, bolt joints are ubiquitously used in multiple industries, such as mechanical engineering, aerospace engineering, and civil engineering. There is an impending need for the periodic inspection and continuous monitoring of bolt looseness, which not only damages the integrity and durability of joints, but also leads to catastrophic consequences [1]. Avoiding bolt self-looseness seems to be almost impossible from both theory and practice, because the actual status of the bolted connections is always associated with the specific service environment, which is recognized as a complex nonlinear system due to various sources of unavoidable uncertainty [2,3]. Hence, it is necessary to explore types of methods to inspect and monitor bolt looseness in a timely manner.

The past few decades have witnessed the development of a number of bolt loosening detecting or monitoring approaches [4,5], including the vibration-based method [6,7], the electro-mechanical impedance (EMI) method [8], the machine vision method [9,10], the ultrasonic-based method, etc. [11,12]. All of them can be divided into various groups in terms of different perspectives, for example, there are active methods and passive methods [13], direct methods and indirect methods [14], offline methods and online methods [15], global methods and local methods [16]. Although these methods have showed remarkable progress on the problem of bolt loosening detection, there are still some existing problems holding back their practical applications. For example, the vibration-based method has proven insensitive to local defects like minor cracks or bolt loosening [6]. The electro-mechanical impedance method is easily affected by environmental factors like temperature fluctuations, which are normally impossible to avoid in practical application [8]. In comparison, the ultrasonic-based method shows great potential in the field of non-damage testing (NDT) [17]. In particular, Wang and Song et al. [12,18,19] achieve encouraging results by using piezoceramic transducers combined with the time-reversion technique. However, ultrasonic instruments are normally high-priced and this kind of method requires the installation of transducers and place circuits, which may lead to a risk of deterioration and violate the user’s original intention to use bolted joints due to their low costs and easy disassembly [20].

As one of the oldest nondestructive testing techniques, the tapping and listening method has been applied in many fields for centuries due to its simple and effective features. It is a low-frequency (less than 1 kHz) elastic wave method based on the transient response of a member to mechanical impact. The surface of the testing workpiece is struck with a metal object such as a steel ball, a hammer, or a heavy chain, which would generate transient waves and set up vibration resonances. The generated wave motion of the surface generates acoustic waves that “leak” into the surrounding air (i.e., acoustic waves) and could be detected by contact sensors mounted on the surface or air-coupled sensors like microphones. The sound produced when a structure is tapped is mainly at the frequencies of the major structural modes of vibration. These modes are structural properties which are related to the local stiffness and damping of the workpiece, so that defects beneath the surface could be detected in this way [21]. Though the names of this method may differ across different fields, the basic principles are identical in nature. For instance, this method is similar to the impact–echo (IE) method for concrete applications and the coin-tap method for composite inspection. Cawley et al. explained the principles of the coin-tap method by using spring theory [22] and Gibson et al. investigated the principles of the impact–echo method based on Lamb wave analysis [23]. It should be stressed that this kind of method is different from the aforementioned vibration method and the wheel-tap method in the railway industry due to their local characteristics. As of now, this kind of method still plays an important role in NDT fields, but some inherent drawbacks should not be overlooked [24,25,26]: (1) it is highly dependent upon the inspector’s hearing and experience; (2) the results are easily subject to interference from background noises; (3) this technique is normally incapable of providing objective data and quantitative information for users; (4) the manual process has weak repeatability and is time-consuming; (5) this technique is limited to some structures with simple geometry. To circumvent the drawbacks above, the human auditory system can be replaced by some portable devices such as smartphones and recorders, which normally have a better response frequency range and sensitivity. These portable devices can also save relatively objective data for later review and analysis [14,27,28,29]. Furthermore, the sound sensing method has been strengthened greatly with pattern recognition techniques and advanced signal processing methods, which could make it possible to provide quantified information for customers and can avoid the interference of background noises to some extent [14,27,29,30]. Researchers and companies have invented some electronic tapping devices, which facilitate the process of detection and promote repeatability [24]. The development of contact mechanics and finite element simulation makes it possible to extend the method to complex structures such as bolted connections [8,26,31,32,33].

Based on precursor investigations, the authors propose a new tapping sound signal processing method, which fuses the time domain and frequency domain together so that it can achieve more promising results. Additionally, we have developed a set of practical sound signal preprocessing algorithms, including denoising, end-point detection and smooth filtering, which can provide relatively standardized signal templates and overcome the effect of ambient noise at a certain level, thus also enhancing the robustness and superiority of the proposed algorithm. The rest of this article is organized as follows. Section 2 introduces the methodologies for audio signal preprocessing and feature extraction from the time and frequency domains, along with a brief description of the generative and discriminative classifier models. Section 3 describes the proposed sound sensing method for bolt loosening detection. Section 4 shows the experimental apparatus and procedures. Section 5 gives the experimental results and analysis. Section 6 concludes the main findings of this paper along with some necessary discussions.

## 2. Methodology

### 2.1. Audio Signal Preprocessing

#### 2.1.1. Denoising

The audio signal recorded by smart devices is normally mixed with a certain level of noise, which will impede the final diagnosis of bolt looseness. Therefore, it is necessary to eliminate or control this kind of influence in order to analyze the signal characteristics. In addition, end-point detection and smooth filtering are also key steps for audio signal processing. For the former, valid samples that could reflect the propagation characteristics of the audio signal are required to be separated from the whole recording by smart devices. For the latter, we expect that the effective samples are continuous and smooth to provide convenience for processing the latter.

As of now, there are several signal denoising methods, such as wavelet denoising, empirical mode decompose (EMD) denoising, and minimum entropy deconvolution (MED) [34]. In this article, a parallel threshold method is adopted. For the signal series x[n],n=1,⋯,N, the threshold h can be set in terms of the level of noise in the original signal series, and then the denoising process can be described in Equation (1).
(1)$$\left.\hat{x}[n]=\left\{\begin{array}{c} x[n]\;|x[n]|-h>0\\ 0\;\;|x[n]|-h\leq 0 \end{array}\right.\right.$$
where *h* indicates the threshold, which can take 0.2 to 0.3 variance in light of the signal–noise ratio (SNR) of the raw signal series (in this paper, we take the threshold as 0.3 variance and the SNR as about 40 dB). The time series Flag(n) can be defined as |x[n]|−h.

The result of a simple comparison between the original signal with white noise and signal after denoising by the parallel threshold method is shown in Figure 1.

#### 2.1.2. Segmenting

In general, the whole recording is to be divided into different frames, which only contain one decay waveform that can be used for feature extraction. An algorithm for audio signal end-point detection is necessary for us, shown in Figure 2. Accurate endpoint detection leads to efficient computation and results in good alignment for template comparison. There are several endpoint detection methods, such as short-time energy (STE), short-time zero crossing rate (ZCR), energy entropy feature, etc. Here, the following steps are implemented to get a relatively satisfying result based on the previous denoising process.

A detection window with a length of L is previously determined and then the speech or silence parts can be identified when the window slides through the whole time series. As for Section 2.1.1, |x[n]|−h is expressed as Flag(n) and then the detection index D(T) is defined as Equation (2):(2)$$D\left(T\right)=\sum_{i=1}^{L}Flag\left(i\right)$$

Here, T=1,2,⋯,N/L and L indicates the number of samples in the window. When the window slides through the whole series, there are four different cases that indicate the status of the speech signal: Case I: D(T)=0 and D(T+1)=0, which indicates the noise part; Case II: D(T)=0 but $D\left(T+1\right)\;\neq \;0$, which represents the starting part and the x(n) corresponding to Flag(L · T) indicates the start point; Case III: $D\left(T\right)\;\neq 0$ and D(T+1) ≠ 0, which shows the speech part; Case IV: $D\left(T\right)\;\neq \;0$ but D(T+1)=0, which indicates the end part and the x(n) corresponding to Flag(T) indicates the end point. Significantly, the key parameter L needs to meet the condition: $d_{1}<L<d_{2}$; $d_{1}$ indicates the minimum interval of silence and $d_{2}$ indicates the maximum interval between the two effective parts.

#### 2.1.3. Smooth Filtering

After the processing of reducing noise and segmenting, raw signals are often discontinuous, which can be solved by smooth filtering. The goal of smooth filtering is to mathematically model the original signal with different fitting curves and enhance its intrinsic characteristics. Similar to blurring in image processing, we can use a quadratic B-spline curve to achieve this goal, which has proven to be simple and effective according to final results. Assuming there are three isolated points $P_{0},\;P_{1}$ and $P_{2}$, the matrix form of the parametric equation of the quadratic B-spline curve can be expressed in Equation (3):(3)$$P(t)=\frac{1}{2}\left[\begin{array}{ccc} t^{2} & t & 1 \end{array}\right]\left[\begin{array}{ccc} 1 & 2 & 1\\ -2 & 2 & 0\\ 1 & 1 & 0 \end{array}\right]\left[\begin{array}{c} P_{0}\\ P_{1}\\ P_{2} \end{array}\right]$$
where t is the parameter of the parametric equation (0≤ t ≤1). Furthermore, a stepwise approach is employed to address the problem of multi-point fitting, i.e., the first quadratic B spline curve is formed by $P_{0},\;P_{1}$ and $P_{2}$, the second quadratic B-spline curve is formed by $P_{1},\;P_{2}$ and $P_{3}\cdots$ and so on. It is relevant to note that boundary processing is to add two extra points $P_{s}$ and $P_{e}$ to the original points, which are required to satisfy the conditions of $P_{s}$ on the extension line where $P_{s}P_{0}=P_{0}P_{1}$ and $P_{e}$ on the extension line where $P_{n-1}P_{n}=P_{n}P_{e}$.

### 2.2. Cumulative Energy Entropy and MFCCs

#### 2.2.1. Cumulative Energy Entropy

It is widely acknowledged that the energy of an audio signal in the time domain can reflect the damage condition of the detection object [35]. However, it has been proven that using a signal energy-based method directly may face some problems, like the damage index (DI) based on energy remaining the same under identical cases [36]. To tackle this problem, the paper proposes a new loosening index based on cumulative energy entropy, which has proven an effective method to tackle the disadvantages mentioned above. Cumulative energy entropy (CEE) can be obtained by the following steps: Firstly, for a given signal that is trimmed by preprocessing: $V_{t}\left(n\right),\;n=1,\cdots ,N$. The energy of a signal can be expressed as shown [29] in Equation (4):(4)$$E(n)=V_{t}^{2}(n)$$

Secondly, the weight of cumulative energy can be computed by Equation (5):(5)$$W(n)=\frac{\sum_{i=1}^{n}\;E(i)}{\sum_{i=1}^{N}\;E(i)}$$

Thirdly, W(n) can be seen as the probability of Shannon entropy [37], and the series of cumulative energy entropy can be expressed by Equation (6):(6)CEE(n)=−W(n)log2 (W(n))

It is noteworthy that the cumulative energy entropy is equal to zero when n is equal to zero and the shape of the CEE curve (CEE versus n) is normally divided into three parts (linear, nonlinear and horizontal segments). The characteristics of the curve can be described by two parameters: the cumulative energy entropy modulus (CEEM) and cumulative energy entropy (CEE) in Equation (7).
(7)$$\left\{\begin{array}{c} CEE=W(N)\\ CEEM=\frac{W(k)}{k} \end{array}\right.$$
where: e indicates a natural number with a value of about 2.714; W(k) indicates the value of cumulative energy entropy corresponding to CEE equal to (1−1/e) CEE; and k is equal to the number of sampling points here.

#### 2.2.2. Mel Frequency Cepstrum Coefficients

Mel frequency cepstrum coefficients (MFCCs) are the most popular parametric representations for acoustic signals. In the MFCC computation process, the speech signal passes through several triangular filters that are spaced linearly in a perceptual mel scale, and the mel filter bank log energy (MFLE) of each filter is calculated. Finally, the cepstral coefficients are computed by using a discrete cosine transformation of MFLE. The specific computing procedure is referred to in the literature [38].

In order to extract the most important features of MFCCs rather than using feature sets with great redundancy, the authors propose a new feature extraction algorithm based on the information gain ratio (IGR) [39]. This algorithm makes an effort to search for the minimum entropy for whole feature sets, which can decrease the order of feature sets obtained from MFCCs and save computational costs with relatively high computational accuracy. The algorithm is divided into two stages: (1) computing the entropy of the original feature sets and the contribution of the information gain ratio for each feature vector; (2) deleting or keeping the feature vector as determined by the IGR.

### 2.3. Machine Learning Techniques: GDA and SVM

#### 2.3.1. Gaussian Discriminant Analysis

Machine learning is the process of creating a set of rules from training data that can then be generalized to the test data [40]. Normally, these techniques can be categorized into supervised learning approaches and unsupervised learning approaches, in light of whether or not the training data are labeled. Furthermore, supervised learning approaches can be specifically divided into discriminative learning algorithms and generative learning algorithms. For the former, these algorithms are trying to learn p(y|x) (the conditional distribution of y given x) directly or to learn mappings directly from the space of inputs x to the labels 0, 1 (such as logistic regression and support vector machines). For the latter, i.e., generative learning algorithms, they are trying to model p(y|x) and p(y) instead.

As one of the generative learning algorithms, Gaussian discriminant analysis (GDA) models the data labels as a Bernoulli distribution and conditional probabilities of the feature vectors as Gaussian distribution [41] as in Equation (8):(8)$$\begin{array}{c} \;\;\;\;y\sim Bernoulli(\phi )\\ x|y=0\sim N(\mu_{0},\Sigma )\\ x|y=1\sim N(\mu_{1},\Sigma ) \end{array}$$

Here, the parameters of the model are $\phi ,\Sigma ,{µ}_{0}$ and ${µ}_{1}$. It is worth noting that though there are two different mean vectors ${µ}_{0}$ and ${µ}_{1}$, this model is usually applied using only one covariance matrix Σ. These parameters are given by the maximum log-likelihood estimates in Equation (9):(9)$$\begin{array}{l} l\left(\phi ,\;\mu_{0},\;\mu_{1},\;\Sigma \right)=log\Pi_{i+1}^{m}p\left(x^{\left(i\right)},\;y^{\left(i\right)};\;\phi ,\;\mu_{0},\;\mu_{1},\;\Sigma \right)\\ =log\Pi_{i+1}^{m}p\left(x^{\left(i\right)}|\;y^{\left(i\right)};\;\phi ,\;\mu_{0},\;\mu_{1},\;\Sigma \right)\\ \times p\left(y^{\left(i\right)};\;\phi \right) \end{array}$$

By maximizing l with respect to the parameters, the unknown distribution parameters can be obtained with Equation (10):(10)$$\begin{array}{l} \phi =\frac{1}{m}\sum_{i=1}^{m}\;I\{y^{\left(i\right)}=1\}\\ \mu_{0}=\frac{\sum_{i=1}^{m}\;I\{y^{\left(i\right)}=0\}x^{\left(i\right)}}{\sum_{i=1}^{m}\;I\{y^{\left(i\right)}=0\}}\\ \mu_{1}=\frac{\sum_{i=1}^{m}\;I\{y^{\left(i\right)}=1\}x^{\left(i\right)}}{\sum_{i=1}^{m}\;I\{y^{\left(i\right)}=1\}}\\ \begin{array}{ccc} \Sigma & = & \frac{1}{m}\sum_{i=1}^{m}\;\left(x^{\left(i\right)}-\mu_{y^{\left(i\right)}}\right){\left(x^{\left(i\right)}-\mu_{y^{\left(i\right)}}\right)}^{T} \end{array} \end{array}$$
where m indicates the number of samples and I(·) represents a logical judgment with a true output of 1, otherwise outputting 0. By applying Bayes’ rule, it can classify a new data point by computing the conditional probability of a class (y= 0 or 1) given the new feature values $x^{\left(h\right)}$. Then, the one with the highest probability will be the predicted class, as in Equation (11):(11)$$y|x^{\left(h\right)}=\left\{\begin{array}{c} 0\;if\;p\left(y=0|x^{\left(h\right)}\right)>p\left(y=1|x^{\left(h\right)}\right)\\ 1\;if\;p\left(y=0|x^{\left(h\right)}\right)<p\left(y=1|x^{\left(h\right)}\right) \end{array}\right.$$

#### 2.3.2. Support Vector Machine

Compared with generative learning approaches, discriminative learning approaches tend to find the optimal classification surface between different categories and to reflect differences between heterogeneous data. Generally, this kind of model outperforms generative learning models because it does not require any assumptions about the datasets. The support vector machine (SVM) algorithm was selected as the audio classification detection algorithm in this article due to its great performance in many fields. A support vector machine can achieve its classification effect by using a hyperplane. As depicted in Figure 3a, a hyperplane w·x+b=0 (represented by the red full line) can separate the data into two classes (negative objects and positive objects), and the margin ∥2/w∥ between the two boundary lines (i.e., w · x+b=±1, represented by the black dotted lines) should be maximized to ensure the best classification accuracy. Similarly, as shown in Figure 3b, we can apply a linear SVM to solve nonlinear classification problems by using a nonlinear mapping function Φ (popular models of kernel include linear, Gaussian, polynomial, etc.). The more specific computing process can be obtained from several articles related to machine learning [30,36] and the authors intend not to repeat them in this paper.

## 3. The Proposed Sound Sensing Method for Bolt Loosening Detection Using GDA and SVM

A schematic view of the proposed sound sensing method for bolt loosening detection is shown in Figure 4. When a hammer strikes bolt connection structures, the sound emission will be transferred to smart devices via microphones, and then the sound file will be analyzed relying on our devised MATLAB 2022b program. In this study, CEE and MFCCs were selected as the damage indexes, and considering the environmental influence in practice, machine learning methods (GDA, SVM) are proposed to conduct deeper classification. After signal preprocessing and feature extraction, the feature vectors, including the time domain (i.e., CEEM and CEE) and frequency domain (i.e., MFCCs) will be classified by GDA and SVM, respectively. Finally, a simple majority vote is utilized for the final judgment of the classifiers. It is worth noting that the audio files that we acquire from smart devices will be divided into two types of datasets: training datasets and test datasets. The GDA and SVM classifier models are constructed using the training datasets, and the test datasets are used to test the performance of the classifier models. To find a method for a cure-all method of loose bolt detection, the experiment was carried out in the laboratory of the Experimental Center of Civil Engineering, with a noise level very close to the real conditions of a construction site.

## 4. Experimental Apparatus and Procedures

To verify the effectiveness of the proposed methods in this paper, a set of repeated experiments were conducted on two steel beams (size: 250 mm × 70 mm × 5 mm, material: Q235) connected with a M8 bolt (class: 8.8, recommended torque: 44 to 58 Nm), as shown in Figure 5. Fixed–fixed boundary conditions were simulated by securing each end of the beam assembly to two screw columns (fastened by double nuts). In light of a previous investigation [42], the results under free–free boundary conditions were similar to those presented here and are not included in this article. Since the proposed methods are based on the tapping and listening method, the experimental setup consists of a contact-type sensor, impact source, and data acquisition/processing/storage system. In this study, a dynamic microphone, which is normally used for music and voice recording, was chosen as the sensor to measure the sound pressure within the near field of the single lap bolt. It should be noted that the distance between the microphone and impact source has a certain influence due to the huge (five orders of magnitude) acoustic mismatch between the steel and air [43,44]. Therefore, this distance was set to approximately 8 cm to account for the size of the workpiece. The response frequency range of the microphone is 20–25,000 Hz and the sensitivity is 103 dB/mW. An ordinary wrench was used to apply an impact to the bolt head and pristine audio signals were recorded and stored in an iPad with internal recording software (sampling rate: 44,100 Hz). In addition, a digital torque wrench and ordinary wrench were utilized for applying axial preloads to the bolt with an interval of 30 Nm. In any case, the levels of ambient noise were about 35 to 45 dB.

In this paper, for convenience, the bolt had three different looseness conditions: fully loosened (0 Nm), partially loosened (30 Nm), and fully tightened (60 Nm). Each condition had 10 datasets and each dataset contained 10 hammer impacts with an interval of about 1 s. Thus, there were 3 × 10 × 10 audio files in total.

## 5. Experimental Research

At each bolt looseness condition, 10 percussions were manually performed using a hammer. The samples of percussion audio signals recorded on a smart device (i.e., iPad) are shown in Figure 6a. The 10 peaks in each plot denote the 10 hammer percussions under each bolt looseness condition. The amplitudes of all peaks are nonuniform because the percussions were manually controlled. As shown in Figure 6b, raw signals in the time domain are typically a decay waveform and the amplitude decreases as the torque applied to the bolt increases. The results are similar to previous studies, where this phenomenon has been interpreted as indicating that more energy is transmitted from the impact source to the microphone, rather than dissipating through bolt structures when the bolt has higher torque [32,45,46].

It is noteworthy that this relationship is not monotonic, because the amplitudes are affected by both the operators and ambient noise, which makes it seem incapable of giving a quantifiable result to denote states of bolt loosening [36]. However, entropy has the characteristics of measuring the complexity and statistical quantification of time series, and thus can tackle the problem mentioned above. Subsequently, signal processing procedures were performed on the raw audio signals. The parameters of the preprocessing algorithms are listed in Table 1 and the classifier results are shown in Figure 7 and Table 2, respectively.

It can be seen from Figure 7 that the CEE curves are basically the same under different conditions, and can be qualitatively divided into three stages: the rapid increase stage, the moderate increase stage and the saturation stage. The duration of the three stages shows an increasing trend. The following three stages can be discussed as follows:

The rapid increase stage (stage I): the CEE has a linear increase approximately correlated with time in this stage, and the growth rate is fast, which indicates that most of the signal energy has a linear decrease in a short time after the occurrence of striking. The moderate increase stage (stage II): the CEE moderately increases in this stage, which means the energy attenuation rate of the received signal gradually decreases and indicates that the signal energy attenuates to a lower level in this stage. The saturating stage (stage III): the CEE reaches its ceiling, and the magnitude tends toward a certain value, which indicates that the sound signal produced by tapping has been completely attenuated, and small fluctuations may be caused by ambient noise.

Comparing the different bolt loosening conditions using their CEE growth curves, it can be seen that the accumulation of entropy under the loosening conditions of 0 Nm and 30 Nm is greater than under the loosening condition of 60 Nm. The straight run of the entropy curve slope can distinguish between the three different bolt looseness states: the tighter the bolt, the steeper the straight segment, the greater the slope, and the CEE will be smaller.

Based on the characteristics of CEE curves discussed above, CEE and CEEM are used to identify the loosening states of bolts in our experiments. As mentioned above, the CEE overall decreases with tighter bolts while the CEEM shows the reverse trend. After being normalized to the whole dataset, this trend became more obvious, as shown in Figure 8.

However, we should admit that the overall differences among cases are not monotonic, especially for partial loosening conditions (torque level: 30 Nm). Nevertheless, it was worth noting that relatively large differences can be observed from the CEE and CEEM results between the fastened conditions and fully loosened conditions, as shown in the Figure 9 (the data were given normalized treatment for ease of viewing).

This phenomenon may be explained through three main reasons: (1) when the bolt was fully loosened, the hammer would cause severe nonlinearity in the received audio signals, which added to fluctuation in the results; (2) the pretension force of the bolt was applied manually and the digital torque wrench itself has a certain error (about ±2%). Therefore, the real torque levels could be more or less than the nominal torque; (3) the results are also limited by the small sample size. After feeding these feature vectors of CEE and CEEM into GDA, the test results can be found in Figure 10.

It can be seen that the GDA model can achieve high training accuracy under different combinations of torque levels. The test results showed that the accuracy rates were 85%, 100%, 95% and 96.7%, respectively, by using the remaining one for cross-validation. Similar to the decision tree method, we could apply an if-elseif structure to classify three different torque levels by two GDA models in series. As shown in Figure 11 and Table 2, the prediction accuracy reached 83.3% under these three different loosening conditions. More specifically, the testing accuracy and testing error, precision, recall values, and F1 measure were also computed (the definitions of these evaluation indexes for the model can be found in reference [41]) as given in Table 2.

In fact, the test error was mainly caused by the fully loosened condition and the partially loosened condition (for the reasons discussed above). Under the combination of these two conditions, the performance of the generative classifier is poorer than under other conditions. Therefore, we could first compute the MFCCs under these conditions. The redundant features among MFCC vectors were deleted in the terms of the IGR algorithm (we obtained MFCC vectors up to 12 orders, whereas only 10 orders of MFCCs were eventually selected for the experiments, as shown in Figure 12).

Finally, we applied SVM to classify them (70% of the data were used to train the model while the rest were used to test). It is worth noting that all of the data were normalized to avoid the influence of the signal amplitude. As shown in Figure 13, the accuracy of our model approaches 100% when parameter γ of rbf (radial basis function $\Phi (x,y)=e^{-\gamma {\left\|x-y\right\|}^{2}}$) is in the range of (0, 2.16). Therefore, a simple majority vote was utilized by weighting the GDA and SVM according to their accuracy on the final judgment of the classifiers in this case (i.e., the combination of full loosening and partial loosening); theoretically, the final accuracy could be improved to 90%.

## 6. Conclusions and Discussion

In this paper, a sound sensing method was developed to further the research on the problem of bolt loosening detection. Firstly, a raw audio signal recorded by smart devices (i.e., iPads) was preprocessed by a series of procedures. Then, new feature vectors were defined in the time and frequency domains. Afterwards, different loosening conditions of bolts were identified automatically by combining a generative learning model (GDA) and a discriminative learning model (SVM). In particular, feature vectors consisting of mel cepstrum frequency coefficients were inputted into SVM to distinguish the fully loosened condition (0 Nm) and partially loosened condition (30 Nm). The experimental results demonstrate that the proposed method could effectively identify bolt looseness (90% for multiple bolt loosening conditions and 96.7% for a combination of a loosening condition and the fully tightened condition). The main findings of this paper are summarized as follows:

(1)Specific preprocessing procedures for audio signals are presented in the paper including denoising, segmenting and smooth filtering. This method enhances the performance of the percussion-based method and can provide standard audio templates for follow-up studies.(2)The concepts of CEE and CEEM are proposed for the first time; they can be viewed as a kind of modified signal energy index to reflect signal characteristics in the time domain. The feature vectors of CEE and CEEM in the time domain and the feature vectors of MFCCs in the frequency domain are recommended for the extraction of bolt loosening indices. Furthermore, a novel feature selection method based on IGR is introduced in this paper.(3)Through the combination of two different supervised learning algorithms, i.e., GDA and SVM, three different torque levels of the bolt were successfully identified and experimental testing results validated the effectiveness and reliability of the proposed method.

The research work in this paper demonstrates the feasibility and superiority of the proposed sound sensing method for bolt looseness detection, by identifying three bolt looseness conditions (0 Nm, 30 Nm and 60 Nm). However, we admit that this work still has some aspects to improve in the future. For example, the experimental results are restrained to small sample sizes (though the GDA and SVM models need less data than other ML models), and because there are multiple paths for sound transmission and reflection; therefore, it is necessary to investigate the influence of different positions of the microphone.

## Figures and Tables

**Figure 1 sensors-24-06447-f001:**
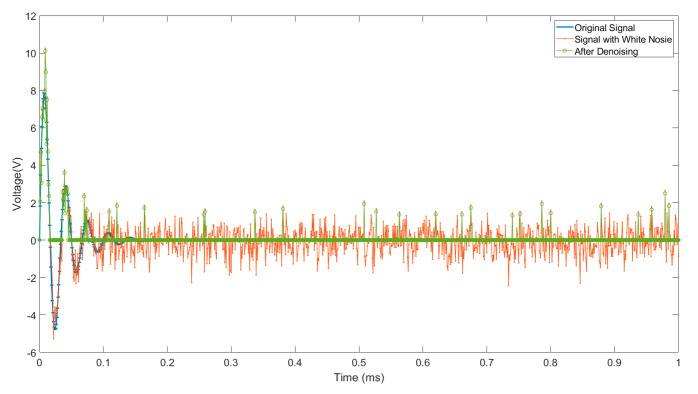
Original signal with white noise and signal after denoising.

**Figure 2 sensors-24-06447-f002:**
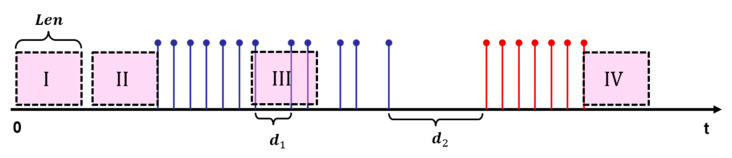
The end-point detection algorithm.

**Figure 3 sensors-24-06447-f003:**
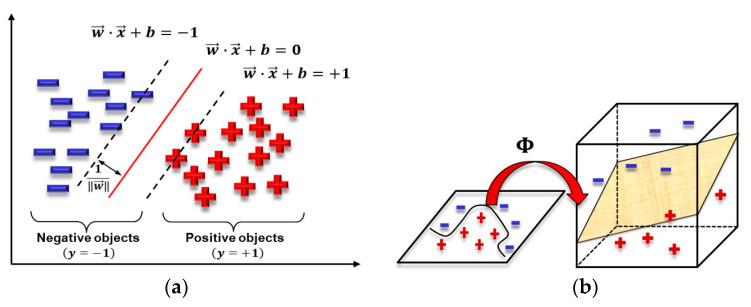
(**a**) SVM for linear classification; (**b**) SVM for nonlinear classification.

**Figure 4 sensors-24-06447-f004:**
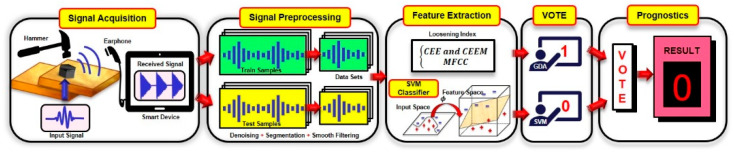
Schematic view of the proposed method.

**Figure 5 sensors-24-06447-f005:**
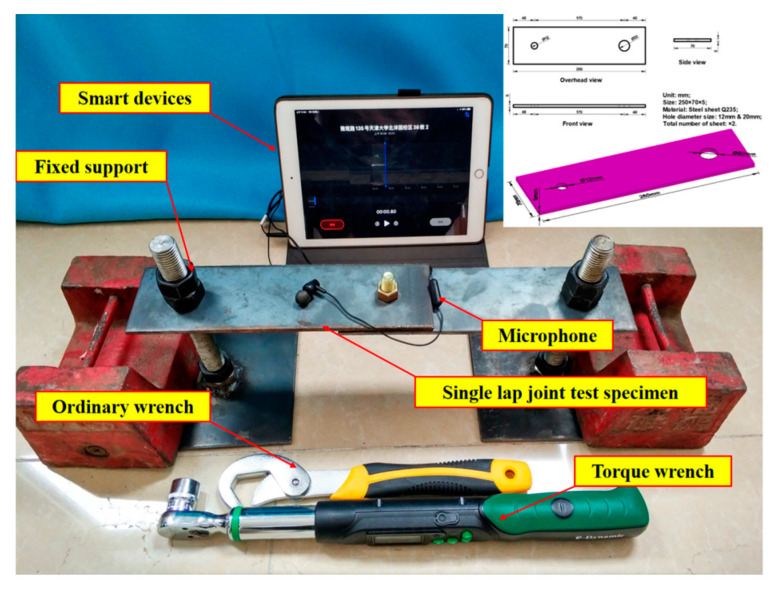
Experimental setup.

**Figure 6 sensors-24-06447-f006:**
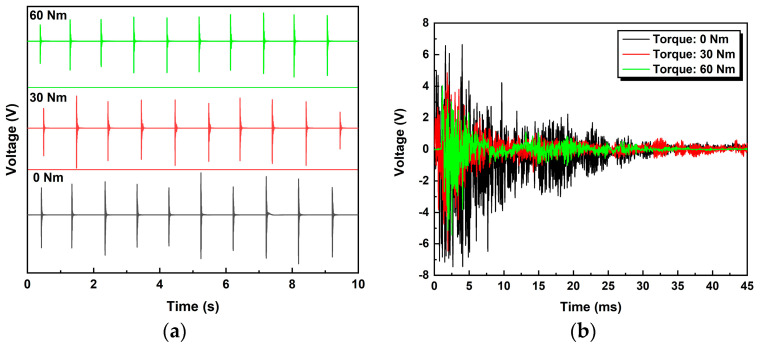
(**a**) Ten impact-induced sound signals for the three bolt looseness conditions; (**b**) raw signals for different torque levels.

**Figure 7 sensors-24-06447-f007:**
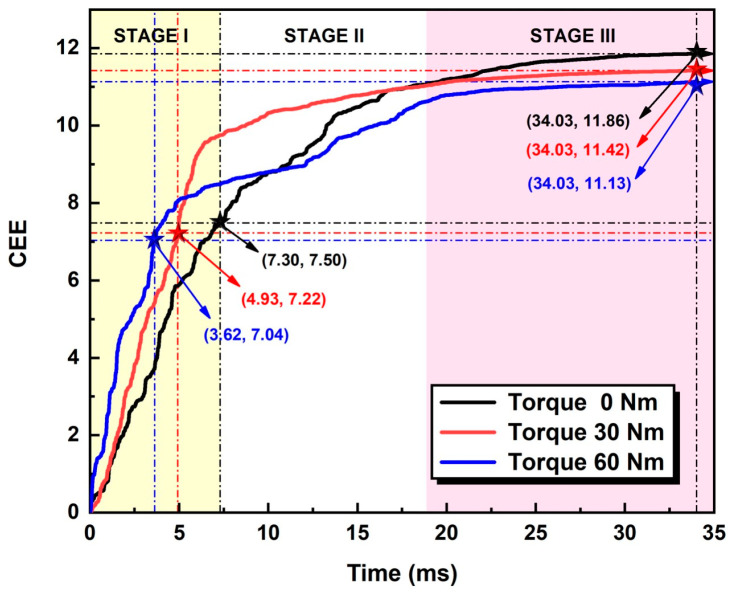
Curves of CEE.

**Figure 8 sensors-24-06447-f008:**
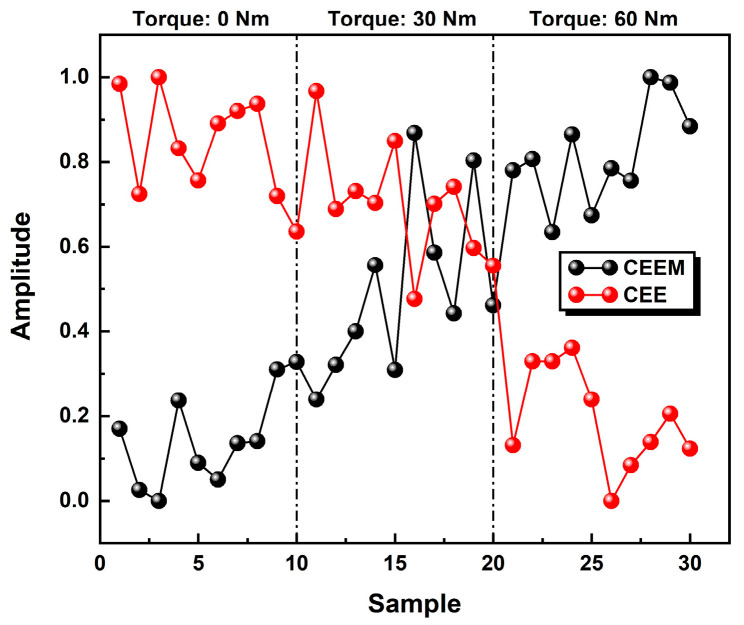
The results of CEE and CEEM under different torque levels for complete datasets (after normalization).

**Figure 9 sensors-24-06447-f009:**
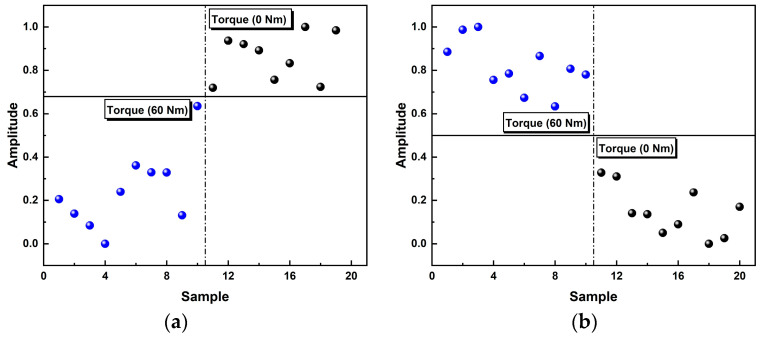
(**a**) CEE results at the torque levels of 0 Nm and 60 Nm; (**b**) CEEM results at the torque levels of 0 Nm and 60 Nm.

**Figure 10 sensors-24-06447-f010:**
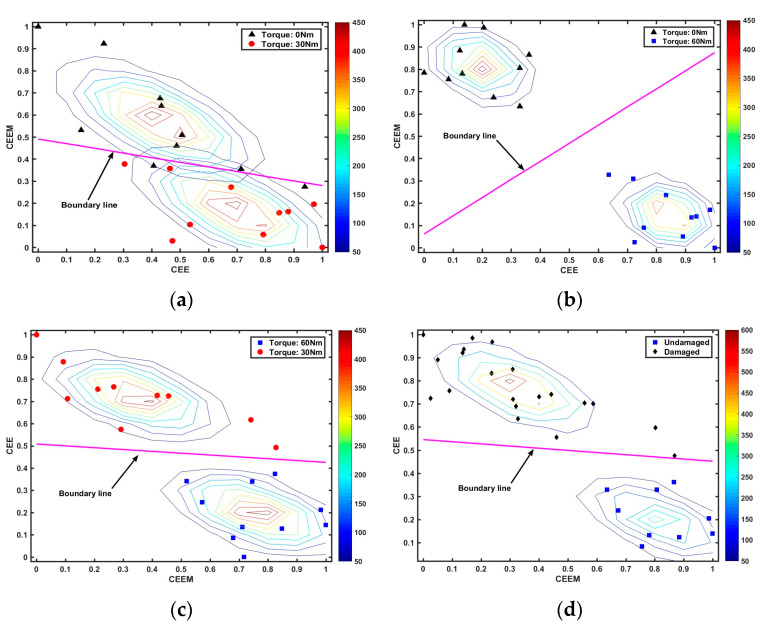
The performance of the GDA model under different combinations of torque levels. (**a**) The performance of the GDA models under the torque levels of 0 Nm and 30 Nm; (**b**) The performance of the GDA models under the torque levels of 0 Nm and 60 Nm; (**c**) The performance of the GDA models under the torque levels of 30 Nm and 60 Nm; (**d**) The performance of the GDA models under the combinations of damaged bolts and undamaged bolts.

**Figure 11 sensors-24-06447-f011:**
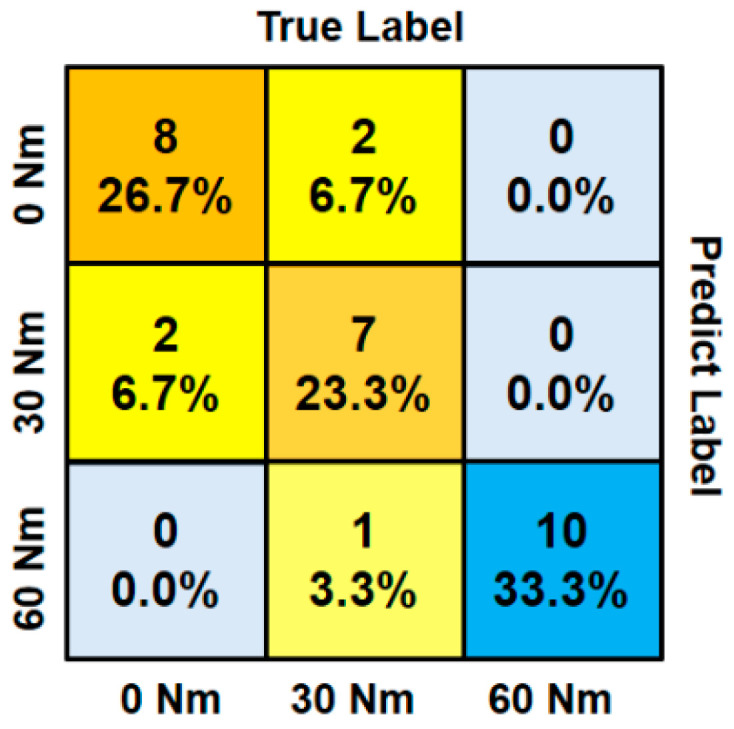
The confusion matrix of the test results.

**Figure 12 sensors-24-06447-f012:**
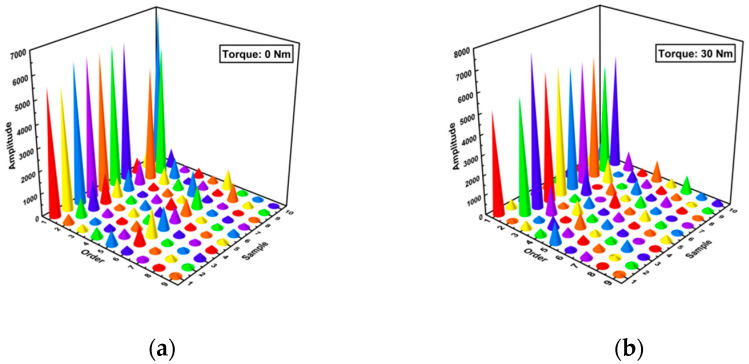
(**a**) The feature vectors using MFCC and IGR under the torque level of 0 Nm; (**b**) The feature vectors using MFCC and IGR under the torque level of 30 Nm.

**Figure 13 sensors-24-06447-f013:**
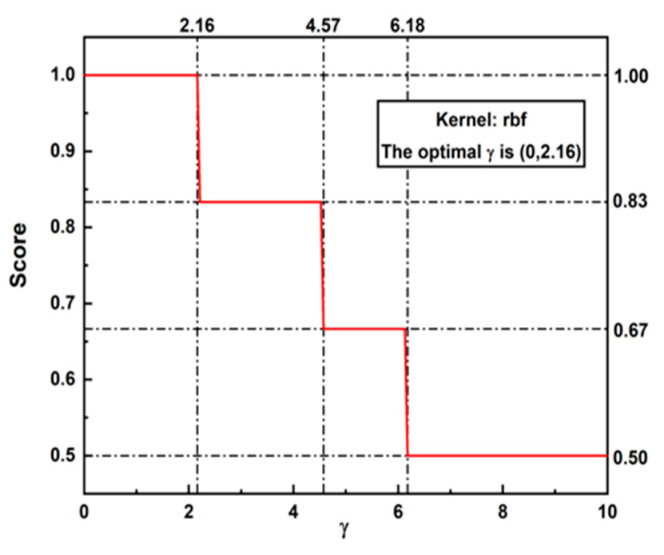
The score curve of SVM under different values of parameter γ.

**Table 1 sensors-24-06447-t001:** Preprocessing parameters.

Stage	Parameter	Value
Denoising	Threshold (h)	0.3
Framing	Length of window (L)	34 ms
Smooth filtering	Order	2
	Number of interpolated points (t)	10

**Table 2 sensors-24-06447-t002:** Model evaluation index.

	*F* _1_	PR	RR	AR	ER
0 Nm	0.80	0.80	0.80		
30 Nm	0.74	0.78	0.70	0.83	0.17
60 Nm	0.95	0.91	1.00		

## Data Availability

Data are contained within the article.

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
