# Peer review of "Sound Sensing: Generative and Discriminant Model-Based Approaches to Bolt Loosening Detection"

_sensors, 2024, doi:10.3390/s24196447_

Round 1

Reviewer 1 Report

Comments and Suggestions for Authors

Authors presented a method of detecting bolt loosening by audio signal processing. The manuscript describes the algorithm and the experiment, showing the performance of the proposed method. The following is this reviewer's comments.

1. '2. Methodology'

Methods are presented without fully explaining why each method is necessary or good for the job.

1) Denoising

1.1) With 40 dB SNR, it seems that denoising is not necessary.

1.2) Is it not clear why the proposed 'parallel threshold method' is good for this problem.

1.3) No change in the wanted signal after the application of the proposed denoising method?

2) Segment

2.1) The logic behind 'the segmentation'. With today's HW/SW technologies, an audio signal of several seconds from an impulse response of a loosened bolt can be processed in its entirety without segmentation.

2.2) The use end-point detection is not clear. The start/end time can be highly accurate (e.g. 10E-8 stability or short-term accuracy with a quartz clock) and the end-point time is known.

3) Smooth filtering

Segmented signal can be continuous since you record the whole audio and segment it digitally by software. So the logic for smooth filtering seems to be weak.

4) CEE, MFCCs, GDA, SVM

4.1) The logic behind the adoption of these techniques is not well explained. Why such a sophistication? Do you try to find a method for a cure-all method of loose bolt detection? But the experiment is done in a laboratory with a simple loose bold configuration.

4.2) Or Are the proposed techniques robust against several loose bolts of different types in the same place?

2. '5. Experimental researches'

1) Experimental verification using a simple setup seems to be not good enough for the proposed method's sophistication. A simple FFT analysis might be as good as the proposed method for this setup.

2) Maybe in-situ real-world loose bolts detection experiments might prove the power of the proposed method.

Author Response

We sincerely appreciate your valuable feedback and recognition of our work. Your insightful comments have greatly contributed to the refinement of our manuscript. We have carefully revised the manuscript in accordance with the suggestions you provided. The detailed response can be founded in the attached file.

Reviewer 2 Report

Comments and Suggestions for Authors

I have carefully reviewed the manuscript titled "Sound sensing: generative and discriminant model-based approaches to bolt loosening detection" submitted to the Sensors journal. The paper presents a novel approach to detect bolt looseness using sound sensing techniques combined with machine learning algorithms, specifically Gaussian Discriminant Analysis (GDA) and Support Vector Machine (SVM). The manuscript is well-structured and addresses an important issue in structural damage detection. However, there are several areas that could be improved to enhance the clarity, impact, and overall quality of the research.

1. This paper proposed a signal preprocessing and denoising method in Section 2.1. Considering that the experiment was carried out indoors, the effect of environmental noise on noise reduction should be discussed. In addition, to verify the necessity and effectiveness of the proposed data processing method, the recognition accuracy without the noise reduction should be calculated during feature extraction.

2. According to Section 3, the authors classified the features using GDA and SVM, respectively. However, most of the experimental results, such as Figure 9, Figure 10, and Table 2, indicated the performance of the GDA method. The classification results and accuracy for the SVM method were not presented. To enhance the integrity and persuasiveness of the research, the authors should add the experimental results for the SVM method as a comparison.

3. This paper used a simple majority vote to obtain the final judgment of the classifiers. However, when the accuracy and performance of the two classifiers (GDA and SVM) differ significantlyfor example, if GDA's accuracy is 90% and SVM's is 70%the accuracy of the majority vote result might be 80%, which is less than the optimal performance of GDA. In such cases, a weighted voting method may perform better than a simple majority vote. The authors should consider this situation and explain the advantages of combining the two methods (GDA and SVM) and how to avoid the potential decrease in recognition accuracy.

Comments on the Quality of English Language

minor editing is needed

Author Response

(The authors gave the same response as above.)
